# Boosting Camera Motion Control for Video Diffusion Transformers

## Abstract

Recent advancements in diffusion models have significantly enhanced the quality of video generation. However, fine-grained control over camera pose remains a challenge. While U-Net-based models have shown promising results for camera control, transformer-based diffusion models (DiT)—the preferred architecture for large-scale video generation—suffer from severe degradation in camera motion accuracy. In this paper, we investigate the underlying causes of this issue and propose solutions tailored to DiT architectures. Our study reveals that camera control performance depends heavily on the choice of conditioning methods rather than camera pose representations that is commonly believed. To address the persistent motion degradation in DiT, we introduce **Camera Motion Guidance (CMG)**, based on classifier-free guidance, which boosts camera control by over 400%. Additionally, we present a sparse camera control pipeline, significantly simplifying the process of specifying camera poses for long videos. Code and models will be released upon publication.

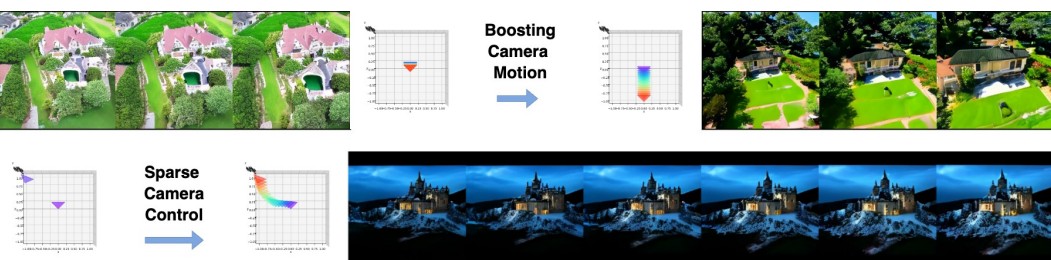

Figure 1: (top) The existing camera control methods implemented for DiT suffer from severe degradation in controllability and produce minimal camera motion. Our methods restore controllability and significantly boost camera motion. (bottom) Our data augmentation pipeline enables smooth camera motion even with sparse camera control, successfully interpolating between only a few or even a single provided camera pose to generate coherent and fluid camera trajectories.

## 1 Introduction

Recent advances in text-to-video (T2V) generation have significantly improved video quality, with diffusion models playing a key role in producing coherent and visually appealing outputs. While these models effectively translate text prompts into dynamic videos, they often lack fine-grained control over aspects like camera pose. To address this, methods such as Wang et al. (2024); He et al. (2024); Xu et al. (2024) introduced camera pose conditioning into U-Net-based (Ronneberger et al. (2015)) diffusion models pretrained on 2D images (Rombach et al. (2022); Podell et al. (2023)), demonstrating promising results in controlling camera trajectories in generated videos. Recently, transformer-based diffusion models (DiT) (Peebles & Xie (2022)) have emerged as the preferred architecture for large-scale video generation models (Ma et al. (2024); Liu et al. (2024); Yang et al. (2024b); Zheng et al. (2024)) due to their superior scalability.

However, existing camera control methods may not be effective for DiT due to the architectural differences. Concurrent work (Bahmani et al. (2024)) found that direct porting of these methods to DiT led to loss of controllability, but no in-depth investigation was conducted to isolate the root

causes. The introduction of new conditioning methods, alongside different camera pose representations, new DiT architecture, and the shift from space-time (2D+1D) to spatio-temporal(3D) latent encoding, further complicated the issue. To address this, we conducted an extensive study to identify the causes of camera control degradation in DiT architectures and proposed solutions broadly applicable across all DiT models and U-Net models.

Our experiments reveal that the strength of camera conditioning weakens in DiT due to the larger embedding dimensions in transformers compared to U-Net. Specifically, the 12-parameter extrinsic camera parameters, a common camera pose representation used in MotionCtrl (Wang et al. (2024)), prove to be ineffective in this context. Although Plücker coordinates used by (He et al. (2024); Xu et al. (2024); Bahmani et al. (2024)) may mitigate the problem slightly but our study reveals it is the camera embedding dimension that plays a more significant role. Improvement can be achieved with appropriate method to project the camera parameters into higher dimension.

Despite these improvements, DiT still experiences significant deterioration in camera motion, even with an optimal combination of camera representation and conditioning methods. Implementing two state-of-the-art U-Net methods (Wang et al. (2024); He et al. (2024)) in DiT resulted in videos with static or limited camera motion and less accurate camera orientations. To address this, we propose a novel Camera Motion Guidance (CMG) method based on the widely used classifier-free guidance technique Ho & Salimans (2021). CMG improves camera pose accuracy and motion over 400% compared to baseline DiT models. Being architecture-agnostic, it applies generically to both space-time and spatio-temporal architecture to improve camera control in video generation.

On the other hand, existing methods for camera control rely on dense camera input, requiring a camera pose for every frame, which is tedious, especially for long videos. To simplify this process, we propose a novel data augmentation pipeline that introduces sparse camera control, where only the camera pose for the final frame or a few key interval frames is required. Our experiments demonstrate that sparse camera control shown promising results, simplifying the input process while maintaining high-quality results and precise camera motion.

We summarize our main contributions as below:

- We introduce novel Camera Motion Guidance, a classifier-free guidance method, improving camera motion by over 400% in video diffusion transformers.

- We identify the root causes of camera control degradation in DiT and successfully developed the first camera control model for space-time video diffusion transformers.

- We develop a novel data augmentation pipeline that enables sparse camera control, which simplifies the required input control signal. To our knowledge, this is a unique feature not demonstrated in the existing work.

## 2 RELATED WORKS

**Video Diffusion Models**. Diffusion models (Ho et al., 2020; Rombach et al., 2022; Stability.ai, 2023; Podell et al., 2023) have achieved remarkable success in image generation, leading to advancements in video diffusion models that build upon this foundation. The video diffusion models (Ho et al., 2022; Blattmann et al., 2023) first process individual frames by applying 2D latent encodings to each image separately, and then fuse the temporal dimension to generate videos, this is known as space-time encoding (2D+1D). More recent developments (Liu et al., 2024; Ma et al., 2024; Zheng et al., 2024; Lab & etc., 2024) have introduced spatio-temporal encoding, which encodes multiple frames simultaneously (3D), creating more compact latent code for long video generation.

**Camera Pose Control** has existed since early 2D human image generation methods. By modifying the size and vertical rotation of skeleton images, methods such as Ma et al. (2017); Ren et al. (2022); Ju et al. (2023); Cheong et al. (2024) could influence camera pose, though only indirectly in limited ways. As neural network architectures have evolved from convolutional networks (Lecun et al., 1998) to transformers (Vaswani et al., 2017), parameterized poses have been explored as substitutes for pose images (Cheong et al., 2022). In a study, Cheong et al. (2023) explicitly mapped 3D camera translation parameters along with 3D SMPL body pose parameters (Lopper et al., 2015), allowing for simultaneous control of both body pose and camera pose.

As video generation emerged as active research, attention has been focused on controlling camera motion for video. AnimateDiff (Guo et al., 2024) trains module on specific motion and hence requiring new module for a different motion, hindering usability. Hence, **parameterized camera control** has recently become a focal point in video generation to improve its ease-of-use. Direct-a-video (Yang et al., 2024a) uses only translation movement limiting the camera motion to only panning and zooming. Instead, MotionCtrl (Wang et al., 2024) introduced a rotation-and-translation matrix derived from camera extrinsic parameters, allowing more complex motion. CameraCtrl (He et al., 2024), on the other hand, uses Plücker coordinates as camera pose representation, enabling geometric interpolation for each pixel. We evaluate both state-of-the-art parameterization methods to assess their effectiveness in DiT-based models. In these methods, camera poses are applied to individual image latents on a frame-by-frame basis, which makes them unsuitable for direct application to spatio-temporal models such as concurrent works (Bahmani et al., 2024; Zhang et al., 2024) where multiple frames are jointly encoded into a single latent representation. To the best of our knowledge, we are the first to develop camera control methods specifically for space-time DiT, enabling video models to leverage the extensive availability of pretrained 2D diffusion models.

**Diffuser Guidance**. Dhariwal & Nichol (2021) demonstrated that applying guidance in diffusion models significantly enhances image quality. During inference, in each denoising step, the model denoises the latent twice—once without conditioning and once with it. The difference between the conditioned and unconditioned latents defines a direction that can be extrapolated by multiplying it with a scalar guidance scale. In particular, classifier-free guidance (Ho & Salimans (2021)) has become ubiquitous in modern text prompting diffusion models in which an empty string or negative text prompt is used as unconditional reference to provide latent extrapolation to improve image or video quality. Despite advancements in camera control for video generation (Wang et al., 2024; He et al., 2024; Xu et al., 2024; Bahmani et al., 2024), diffuser guidance has not been explored for camera motion improvement. In this paper, we introduce camera motion guidance to enhance the accuracy and quality of camera motion in video generation models.

**Sparse Camera Control.** Existing methods rely on dense camera poses to achieve effective control. SparseCtrl (Guo et al., 2023) explores applying sparse image-based structural control but does not incorporate camera control, leaving a gap in addressing sparse camera pose scenarios for video generation tasks.

# 3 OUR APPROACH

## 3.1 PRELIMINARY

**Camera Representation.** Extrinsic camera parameters describe the camera's position and orientation in 3D space represented by a rotation matrix $\mathbf{R} \in \mathbb{R}^{3\times3}$ and a translation vector $\mathbf{T} \in \mathbb{R}^3$ which form the rotation-and-translation (RT) matrix $[\mathbf{R}|\mathbf{T}] \in \mathbb{R}^{3\times4}$. Intrinsic parameters, encapsulated in the camera matrix $\mathbf{K}$, define the camera's internal characteristics, including focal length, principal point and pixel size. These are used to map a 2D pixel location in the image to a 3D direction vector in camera's coordinate system. For each pixel $(x, y)$ in image coordinate space, its Plücker coordinate is calculated as $(\mathbf{O} \times \mathbf{d}_{x,y}, \mathbf{d}_{x,y})$ where $\mathbf{O} \in \mathbb{R}^3$ is the camera center in world coordinates derived from $-\mathbf{R}^\top \mathbf{T}$; and the direction vector $\mathbf{d} \in \mathbb{R}^3$ is obtained by:

$$\mathbf{d}_{x,y} = \mathbf{R}\mathbf{K}^{-1}[x, y, 1]^\top \tag{1}$$

This formulation represents the direction and location of 3D lines, allowing for efficient geometric operations like interpolation and transformation, which are useful for camera trajectory and ray-based rendering. Compared to a flattened RT $\in \mathbb{R}^{12}$ in a video frame, Plücker coordinates has higher dimension $\in \mathbb{R}^{h,w,6}$ where $h$ and $w$ are height and width of video resolution.

**Camera Conditioning for Video Generation.** We examine the state-of-the-art camera conditioning in U-Net models to understand how their network topology differences from DiT models affect the effectiveness of camera conditioning. In U-Net architecture, the spatial resolution decreases as the features traverses down the network, while the channel $C$ increases from e.g. 320 to a maximum of e.g. 1280 before stepping down again. MotionCtrl(Wang et al. (2024)) employs a RT matrix to as camera pose representation for the entire video frame. To incorporate this into the U-Net, the flattened RT is repeated for every latent pixel and concatenated with the U-Net's features along channel dimension, forming a new dimension of $C+12$ where $C$ is the U-Net embedding's channel

number. However, increased spatial resolution of U-Net's embedding is accompanied with repetitive RT in each spatial position and does not carry more camera information. Thus we can ignore the spatial dimension when considering camera conditioning influence and consider only the channel dimension. As illustrated in Figure 2(a), U-Net has the smallest embedding channel at its top and bottom layers, resulting in the highest ratio of camera parameter dimension to embedding dimension, we refer to as *condition-to-channel ratio*. For simplicity, we display only the channel dimension of the full tensor shape $[B, N, H, W, C]$ (batch size, number of frames, image height, image width, channel) and omit the other dimensions in the figure.

In contrast, transformers maintain a consistent channel number e.g. 1024 across all the layers. When implementing MotionCtrl's method to OpenSora (Zheng et al. (2024)) DiT, the increase of minimum channel number from 320 to 1152 lead to significant drop ($3\times$) of the condition-to-channel ratio of 12:$C$ from 1:27 to 1:96. We hypothesize camera conditioning strength is proportionate to this ratio, which leads to a considerably weakened control strength in DiT.

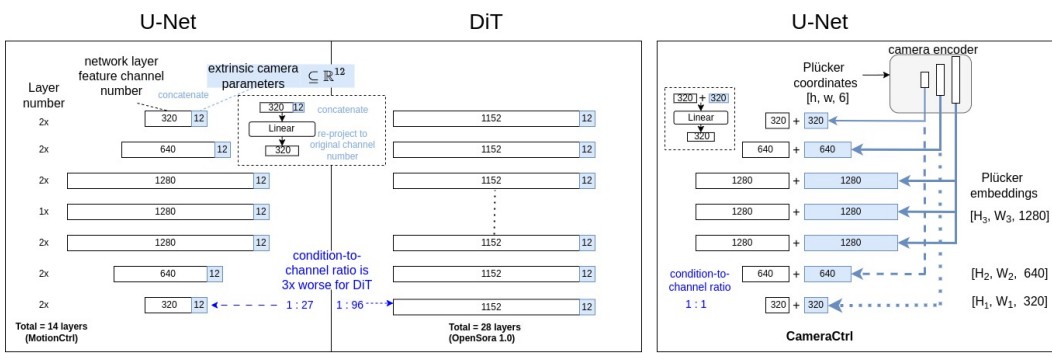

(a) MotionCtrl Method (RT, concatenation)     (b) CameraCtrl Method (Plücker, add)

Figure 2: We illustrate two prominent camera control U-Net model architecture (a) concatenation of rotational-and-translation matrix, and (b) addition of Plücker embedding. For clarity, only channel dimension is shown, omitting batch, frame, height and width of features. Comparing with U-Net, DiT has a worse condition-to-channel ratio leading to weakened control.

Our experiment compares another conditioning scheme from CameraCtrl (He et al. (2024)) to confirm our hypothesis. As Plücker coordinates has dimension of $[h, l, 6]$ per camera pose, a camera encoder is necessary to produce multi-scale camera embedding that matches the dimensions of the DiT's features in each layer $l$, represented as $[H_l, H_l, C_l]$ for element-wise addition as shown in Figure 2b. This difference is even more pronounced when considering the spatial dimension, as each Plücker embedding is unique for each spatial location, thereby offering significantly stronger camera conditioning. At a high level, the main differentiators between the methods are the camera representations and the approach to fusing camera embeddings: MotionCtrl concatenates RT, while CameraCtrl adds Plücker embedding to the U-Net features.

Both methods share several common practices. First, each fused embedding is projected back to the original embedding dimension via a linear layer at every U-Net layer. Additionally, camera conditioning is applied before the temporal transformers. Only the temporal transformers and the newly added camera control components are updated during training, while all other parameters remain frozen. These practices were similarly adopted in our experiments. Another method, CamCo (Xu et al. (2024)), employs a conditioning scheme similar to CameraCtrl, with the key difference being the use of 1×1 convolution layers instead of linear layers for projection. However, due to the lack of open-source implementation, we have not experimented with it in our study.

## 3.2 DiT Baseline Model Implementation

We adopts an open source DiT text-to-video (T2V) model OpenSora (Zheng et al. (2024)) as our base video generation model. They have newer versions that employ a spatio-temporal 3D encoder, enabling higher resolution and better image quality. However, we use OpenSora 1.0 that uses space-time architecture, allowing for direct comparison with the U-Net models. Then OpenSora 1.0 model

consists of a cascade of 28 Spatial-Temporal DiT (ST-DiT) blocks inspired by Ma et al. (2024), each containing a spatial transformer followed by a temporal transformer.

We implemented two methods as our DiT baselines: DiT-MotionCtrl and DiT-CameraCtrl, named after their respective U-Net-based counterparts. The methods were faithfully implemented for like-for-like comparison, with adjustments to the channel dimension of linear layers and the camera encoder to match the embedding dimension of DiT. Additionally, given DiT's uniform channel dimension, we have only one block in the camera encoder, in contrast to three blocks in U-Net's producing camera embeddings at three different resolutions.

### 3.3 Data Processing and Camera Augmentation

Given that our base video model is limited to 16 frames, translating to about 0.5 seconds of footage at 30 frames-per-second, taking consecutive frames would result in minimal camera motion. Therefore, we extract video frames from the dataset in strides of 4 to 8 frames, sampled uniformly, ensuring we capture sufficient temporal information while preserving meaningful motion dynamics. We randomly sample 16 frames from a training video sample, and this can produce arbitrary large starting translation vector. Therefore, we represent all RT matrices relative to the first frame by setting the translation vector $\mathbf{T}_1 = \mathbf{0}_3$ and rotation matrix $\mathbf{R}_1 = \mathbf{I}_{3\times3}$ and multiplying $[\mathbf{R}_1|\mathbf{T}_1]^{-1}$ to the rest of the RT matrices.

To ensure smooth integration of our novel camera control method, we now establish a robust camera augmentation pipeline. In training, we randomly generate *static video* by repeating the first frame and its corresponding camera pose across all subsequent frames. In other words, all camera poses are filled with value of $[\mathbf{R}_1|\mathbf{T}_1]$ or its corresponding Plücker coordinate, a condition we denote as *null camera*, $\emptyset_C$. Inspired by Srivastava et al. (2014), we also randomly drop out camera poses in a video by setting them to zeros to prevent overfitting and allow for sparse camera control, we call this *zero camera*. In addition to the new proposed augmentation, we also adopted standard video augmentation of center image cropping and video temporal reversal which is critical as it balances the distribution of two opposite camera motions.

### 3.4 Camera Motion Guidance (CMG)

We now introduce our novel method - Camera Motion Guidance- based on classifier-free guidance. Equation 2 shows the generic classifier-free guidance equation for text prompt (Ho & Salimans (2021)).

$$\hat{e}_\theta(z_t, C_T) = e_\theta(z_t, \emptyset_T) + s_T\{e_\theta(z_t, C_T) - e_\theta(z_t, \emptyset_T)\} \tag{2}$$

where $e_\theta$ is the denoising model (U-Net or DiT), $z_t$ is the noisy latent at time step $t$, $C_T$ is the text condition, $\emptyset_T$ is null text condition and $s_T$ is text prompt guidance scale. In essence, the second term in the equation finds the text embedding direction in the latent space and extrapolates it with a scalar $s_T$, using the unconditioned first term as a reference point.

In existing camera-controlling literature (Wang et al., 2024; He et al., 2024), classifier-free guidance is applied solely to the text prompt, with the camera condition $C_C$ present in all guidance terms. This approach is analogous to Equation 3, effectively canceling out the camera condition's influence in the second term.

$$\hat{e}_\theta(z_t, C_T, C_C) = e_\theta(z_t, \emptyset_T, C_C) + s_T\{e_\theta(z_t, C_T, C_C) - e_\theta(z_t, \emptyset_T, C_C)\} \tag{3}$$

To address this, we propose a new camera motion guidance term, disentangling it from the original text guidance term, as outlined in Equation 4. The null camera $\emptyset_C$ is used as a reference, while the camera motion guidance scale, $s_C$, is applied to guide the camera motion independently.

$$\begin{aligned}\hat{e}_\theta(z_t, C_T, C_C) = {}& e_\theta(z_t, \emptyset_T, \emptyset_C) + s_T\{e_\theta(z_t, C_T, \emptyset_C) - e_\theta(z_t, \emptyset_T, \emptyset_C)\} \\ & + s_C\{e_\theta(z_t, C_T, C_C) - e_\theta(z_t, C_T, \emptyset_C)\}\end{aligned} \tag{4}$$

With simple changes in data augmentation, our method CMG can enhance any video generative model employing classifier-free guidance, offering improved camera control across a variety of architectures.

## 4 EXPERIMENTS

### 4.1 IMPLEMENTATION DETAILS

We train our models using the RealEstate10k dataset (Zhou et al., 2018), which features indoor and outdoor real estate videos with corresponding camera poses. The models are trained at a resolution of 256×256 and 16 frames per video sample, matching the settings of the U-Net models for comparisons. Since the original dataset lacks captions, we use the text prompts provided by He et al. (2024). The training was conducted on GPUs with 40GB memory, using a batch size of 3 per GPU, and models were trained for 8 epochs with a fixed learning rate of $1 \times 10^{-5}$.

We evaluate two baseline DiT models: DiT-MotionCtrl and DiT-CameraCtrl, which use RT matrices and Plücker coordinates as their respective camera representations. Additionally, to enable CMG, we train the same models with our augmentation pipeline which saw 5% null camera data augmentation. Both the baseline and CMG versions also applied a 5% camera dropout, which is randomly set between 70% and 100% of camera frames in a video to zero.

During inference, the two baseline models use standard guidance (Eq. 3), while CMG models apply Eq. 4. A text guidance scale $s_T$ of 4.0 is used consistently across all the DiT models. We evaluate a range of CMG scale $s_C$ between 4.0 and 7.0, eventually selecting 5.0 for comparison with the baselines. Apart from this, identical configurations and random seeds are applied to all DiT models to ensure fair comparisons. We use U-Net models' default configurations for inferences. To investigate the effect of camera representation further, we also replace the Plücker coordinates in DiT-CameraCtrl with RT matrices and re-train a new model.

### 4.2 METRICS

We aim to measure two aspects of camera motion: first, the accuracy with which the camera motion adheres to the specified camera conditioning and second, the extent of motion present in the generated videos. For the camera motion accuracy, we adopt approach from He et al. (2024) to utilize COLMAP (Schönberger & Frahm, 2016) in extracting rotation matrices $\mathbf{R}_{gen} \in \mathbb{R}^{N \times 3 \times 3}$ and translation vectors $\mathbf{T}_{gen} \in \mathbb{R}^{N \times 3}$ of generated videos where $N$ is frame length. The rotation error $\mathbf{R}_{err}$ and translation error $\mathbf{T}_{err}$ are calculated by comparing with ground truth $\mathbf{R}_{gt}$ and $\mathbf{T}_{gt}$ respectively using Eq. 5 and 6.

$$\mathbf{R}_{err} = \sum_{n=2}^{N} \cos^{-1}\left(\frac{tr(\mathbf{R}^n{}_{gen}\mathbf{R}_{gt}^{n\top}) - 1}{2}\right) \tag{5}$$

$$\mathbf{T}_{err} = \sum_{n=2}^{N} \|\mathbf{T}_{gt}^n - \mathbf{T}_{gen}^n\|_2 \tag{6}$$

where $n$ is $n$-th video frame and $tr$ is the trace of a matrix. We exclude the calculation of error for the first frame, as it is always zero by definition due to the camera poses preprocessing. We report the rotation error in radian. The translation range in generated videos can vary, but more importantly, COLMAP estimation can yield a wide translation range. Therefore, we normalize both translation vectors $\mathbf{T}_{gt}$ and $\mathbf{T}_{gen}$ to have a unit maximum distance during inference for metric evaluation.

To quantitatively assess the level of motion in the generated videos, it is essential to identify an appropriate measurement method. After considering various options, we ultimately select Teed & Deng (2020) to measure the optical flow between two frames, which gives a flow field represented as two arrays $u$ and $v$ corresponding to each pixel's horizontal and vertical components of motion. The motion magnitude $M$ is then calculated for all adjacent frames using Eq. 7, and the results are averaged over the entire video.

$$M = \frac{1}{K} \sum_{k=1}^{K} \sqrt{u_k^2 + v_k^2} \tag{7}$$

where $k$ is $k$-th pixel in a video frame and $K$ is total pixel counts in a frame. We also use FID (Heusel et al. (2017)) to measure image quality in the ablation study, ensuring that our method maintains high video quality.

## 4.3 RESULTS

**Motion Degradation with DiT Implementation**. Porting the MotionCtrl method into DiT leads to a total loss of controllability, as also observed by Bahmani et al. (2024). This is evident in sharp rise in rotation error as shown in Table 1 (Model 1a→1b). Most importantly, the motion magnitude collapses 80% from 7.780 to 1.485. The lack of motion is difficult to perceive from still images, therefore we included "Supplementary Video 1 - Method Comparison" to demonstrate the stark contrast in motion. Although applying CMG to DiT-MotionCtrl (Model 1c) brings slight improvement(compared to Model 1b), the high errors and the lack of motion persist, indicating the corresponding U-Net method is not effective for DiT.

| Model | Condition-to-Channel Ratio (CCR) ↑ | Camera | RotErr ↓ | TransErr ↓ | Motion ↑ |
|---|---|---|---|---|---|
| (1a) MotionCtrl (U-Net) | 1:27 | RT | 0.168 | 0.640 | 7.786 |
| (2a) CameraCtrl (U-Net) | **1: 1** | Plücker | 0.176 | 0.754 | 9.686 |
| (1b) DiT-MotionCtrl (baseline) | 1:96 | RT | 0.224 | 0.716 | 1.485 |
| (1c) DiT-MotionCtrl w CMG (Ours) | 1:96 | RT | 0.208 | 0.702 | 1.806 |
| (2b) DiT-CameraCtrl (baseline) | **1: 1** | Plücker | 0.186 | 0.687 | 1.564 |
| (2c) DiT-CameraCtrl w CMG (Ours) | **1: 1** | Plücker | **0.176** | **0.577** | **6.450** |

Table 1: Quantitative results showing the performance degradation of camera conditioning implemented for DiT architecture. Our method of applying CMG to DiT-CameraCtrl significantly improves the metrics, outperforming all DiT baselines.

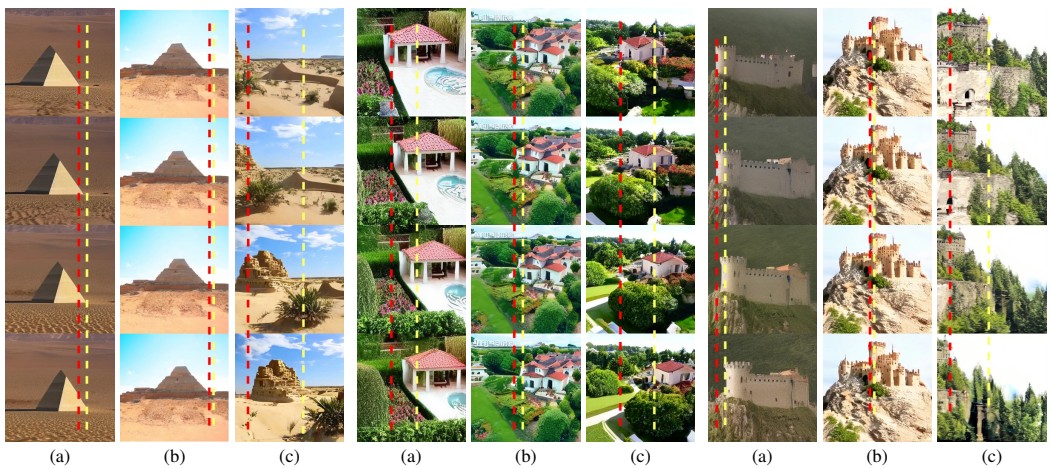

(a)    (b)    (c)        (a)    (b)    (c)        (a)    (b)    (c)

Figure 3: (a) CameraCtrl (b) DiT-CameraCtrl (c) DiT-CameraCtrl w CMG (Ours). Direct DiT implementation of CameraCtrl results in severe loss of camera motion in specified pan left/right camera condition. However, applying our method CMG restore the controllability and boosting the camera motion.

**Camera Motion Guidance Restores Controllability and Significantly Boosting Motion.** While the baseline DiT-CameraCtrl's rotation error remained at a similar level, meaning it has better controllability, it still suffered from severe losses in motion and translation accuracy (Model 2a→2b) as illustrated in Figure 3 and "Supplementary Video 1 - Method Comparison". Applying CMG significantly boosted both metrics, with a notable 412% increase in motion magnitude, from 1.565 to 6.450 (Model 2b→2c). Our model (Model 2c) significantly outperformed both DiT baseline models (Model 1b and 2b) and also surpassed both U-Net models in translation error, which is a more critical metric for our study than rotation error. It is worth noting that, motion as measured from optical flow is sensitive to pixel quality and video content. Therefore, the motion magnitude is not directly comparable with U-Net models which are based on different pre-trained base models and training recipes.

**Comparing Plücker Coordinates and Extrinsic Parameters**. Previously in Table 1, we have shown that DiT-CameraCtrl (Model 2b) is more effective than DiT-MotionCtrl (Model 1b), and the

performance gap becomes even more significant with our CMG (Model 1c→ 2c). However, aside from their conditioning methods, the two methods also use different camera data representation. To make a fair comparison, we repeated the DiT-CameraCtrl experiment by replacing the Plücker coordinates with RT matrices. This allowed us to isolate and compare the methods with only one differentiating factor at a time.

| Model | Camera | CCR ↑ | RotErr ↓ | TransErr ↓ | Motion ↑ |
|---|---|---|---|---|---|
| (1b) DiT-MotionCtrl | RT | 1:96 | 0.224 | 0.716 | 1.485 |
| (2b) DiT-CameraCtrl | Plücker | **1:1** | 0.186 | **0.687** | 1.564 |
| (2d) DiT-CameraCtrl | RT | **1:1** | **0.177** | 0.748 | **2.101** |
| (2c) DiT-CameraCtrl w CMG | Plücker | **1:1** | **0.176** | **0.577** | **6.450** |
| (2e) DiT-CameraCtrl w CMG | RT | **1:1** | 0.177 | 0.666 | 5.721 |

Table 2: Comparing individual effect of model and camera representation. Results for Model 1b, 2b, 2c are included from Table 1 for ease of comparison.

From Table 2, we can draw three key insights: first, RT is not inherently inferior, as DiT-CameraCtrl achieves better overall results with RT (Model 2d) compared to Plücker coordinate (Model 2b). Secondly, conditioning method plays a crucial role. DiT-CameraCtrl consistently outperforms DiT-MotionCtrl for both camera representation (Model 2b,d vs 1b), confirming our hypothesis that a better condition-to-channel ratio (CCR) strengthens camera conditioning. Lastly, our CMG method consistently boosted DiT-CameraCtrl's performance in both Plücker coordinate (Model 2b→2c) and RT (Model 2d→ 2e), and also improve DiT-MotionCtrl (Table 1:1b→1c). This demonstrates the robustness and effectiveness of CMG in improving camera control across different configurations.

## 4.4 ABLATIONS

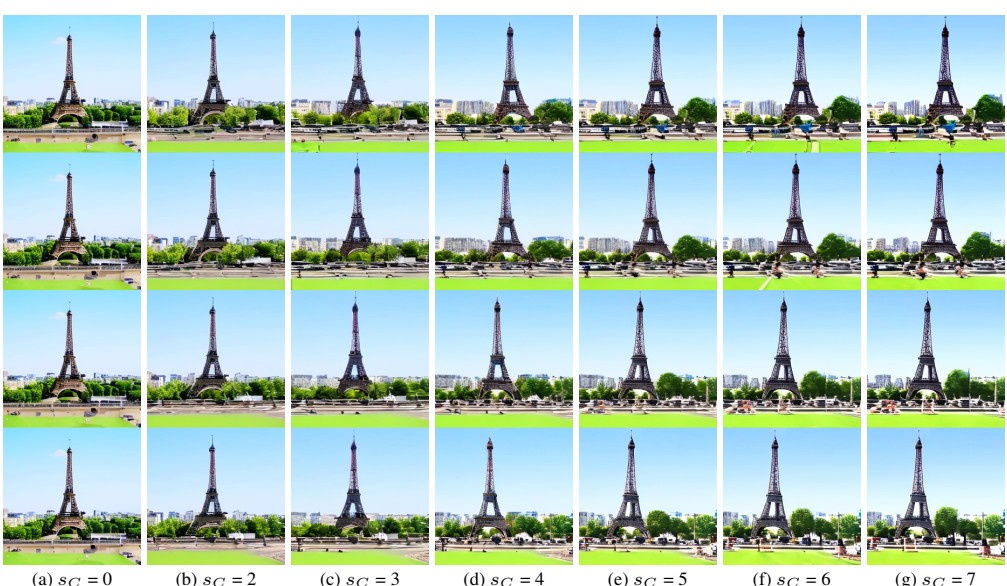

| (a) $s_C = 0$ | (b) $s_C = 2$ | (c) $s_C = 3$ | (d) $s_C = 4$ | (e) $s_C = 5$ | (f) $s_C = 6$ | (g) $s_C = 7$ |

Figure 4: Each column is showing a video generated with a different CMG scale $s_C$. As the scaling increases, the specified camera motion (pan right) is also more pronounced, evident from greater translation in the Eiffel Tower's position. The visually similar videos also demonstrate effective disentanglement of our camera motion guidance from the text guidance term.

We conduct an extensive study varying the CMG scale across a range of values to prove its effectiveness in inducing and controlling camera motion. As illustrated in Figure 4, increasing the CMG scale results in greater camera motion. The preservation of video content highlights the strong disentanglement of our method, allowing independent camera motion control, separated from the text guidance in classifier-free guidance (Eq. 4) that describes the video scene.

As DiT-MotionCtrl has proven ineffective, we focus our discussion on DiT-CameraCtrl for the quantitative result shown in Table 3. Determining the optimal CMG scale is not straightforward, as no single value consistently delivers the best results across all metrics. Additionally, each metric has its

| | Rot error ↓ | Transl error ↓ | Motion ↑ | FID ↓ (vs CameraCtrl) |
|---|---|---|---|---|
| DiT-CameraCtrl | 0.186 | 0.687 | 1.564 | 91.3 |
| DiT-CameraCtrl w CMG= 4 | 0.172 | 0.595 | 5.721 | **69.5** |
| DiT-CameraCtrl w CMG= 5 | 0.176 | **0.577** | 6.450 | 70.5 |
| DiT-CameraCtrl w CMG= 6 | 0.180 | 0.597 | 7.061 | 71.3 |
| DiT-CameraCtrl w CMG= 7 | **0.169** | 0.600 | **7.631** | 73.8 |

Table 3: Ablation of different CMG scales.

own limitations, making the selection of an ideal CMG scale more nuanced. While higher motion magnitude increases movement, it can also result in blurrier videos, as reflected by the degradation in FID scores compared to those produced by U-Net model using the same text prompt. Among the error measurements, translation error plays a more critical role in typical camera motion scenarios. Therefore, we selected CMG scale of 5.0 which minimizes translation error for optimal performance, and used it for main comparison in Table 1.

### 4.5 SPARSE CAMERA CONTROL

Since we drop certain interval frames by setting the camera poses to zeros during training, our method allows users to provide camera control for only a sparse set of frames at test time, which, to our knowledge, is not supported by existing methods. Translation motion in a single dimension, such as zooming, can be easily interpolated and does not offer significant value in testing sparse control. Therefore, we excluded simple translation motion from evaluation. Table 4 presents the rotation and translation errors at different sparsity ratios, which we define as the ratio of dropped camera poses to $N - 1$ frames, excluding the first frame. While errors do increase with higher sparsity ratios, the rate of error increase remains relatively modest compared to the level of sparsity, even up to 87% sparsity, where only the camera poses in the first, middle, and last frames are provided.

| # camera poses dropped | Sparsity Ratio | Rot error ↓ | Transl error ↓ |
|---|---|---|---|
| 0 | 0% | **0.176** | **0.611** |
| 7 | 47% | 0.181 | 0.627 |
| 11 | 73% | 0.203 | 0.650 |
| 13 | 87% | 0.218 | 0.755 |
| 14 | 93% | 0.291 | 0.768 |

Table 4: Increasing camera control sparsity results only in modest drop in controllability as measure dby rotation and translation error.

Figure 5 shows videos generated using sparse camera poses as specified in the leftmost column where only 4 frames (73% sparsity) and 1 frame (93% sparsity) are used respectively. When only the last frame was used, the specified camera pose Figure 5a and Figure 5b end in similar position, differing only in the camera rotation. In Figure 5a, the generated camera motion accurately follows the translation-only trajectory, maintaining a straight-facing camera angle as expected. On the other hand, the rotated ending camera pose in Figure 5b result in a smooth rotating motion alongside translation, similar to the video above generated using denser camera poses. The videos, which can be viewed in "Supplementary Video 3: Sparse Control" highlights our model's ability to interpolate the camera poses to fill in the gaps and maintaining smooth, coherent camera motion. Our sparse camera data augmentation technique is also effective with standard DiT methods without CMG. While this model demonstrates weaker camera controllability and motion without CMG, it still successfully interpolates camera poses, proving its robustness in sparse control scenarios.

## 5 LIMITATIONS

Although object motion is excluded from our study, it is not negatively impacted by CMG. In "Supplementary Video 4 - Object Motion", we showcase object motion from natural landscapes and 3D character animation alongside camera motion controlled using our method. However, the videos generated by our models may show limitations in image quality and content richness compared to models pre-trained on larger datasets. The OpenSora 1.0 we use was pre-trained on 400K video clips—a much smaller dataset than the 10M videos used by MotionCtrl. This constraint may also have led to occasional deformations for objects such as the Eiffel Tower, which are not present in the RealEstate10k dataset we trained on. Since our CMG method effectively disentangles camera

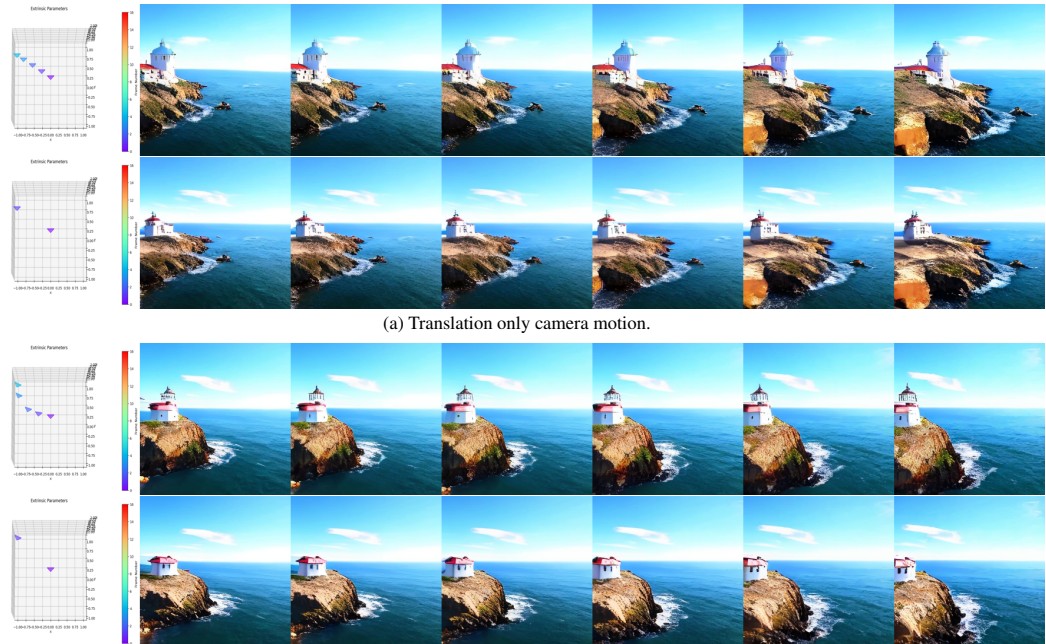

(a) Translation only camera motion.

(b) Simultaneous translation and rotation camera motion.

Figure 5: Videos following specified camera trajectories with sparse camera control as shown in the left.

motion from the text prompt, we believe more visually appealing videos could be generated with a higher-quality base video model.

## 6 CONCLUSIONS

This paper thoroughly examined the impact of various camera representations and conditioning methods on camera control for video generative diffusion transformers. Our extensive experiments confirmed that high-dimensional camera embedding is critical for effective camera control, supporting our hypothesis. We also found that camera representation alone is not the key factor for successful control; instead, it must be paired with effective conditioning methods and guidance techniques to achieve optimal results. We successfully demonstrated the first camera control model for spacetime DiT by combining the CameraCtrl architecture, Plücker coordinates for camera representation, and our novel camera motion guidance (CMG).

We have proved that CMG is highly effective in inducing motion and enhancing camera control. Due to limited resources and code availability, we could not experiment with CMG on a broader range of video models. However, we believe it would be equally effective for U-Net and other DiT models with spatio-temporal architectures. Additionally, we introduced novel camera data augmentation techniques that enable sparse camera control. These simple yet effective methods are generic, making them applicable and beneficial for a broader range of video model architectures.

In future work, we aim to test our CMG method on a wider range of models, including U-Nets and other spatio-temporal DiT. Additionally, we plan to enhance our approach to sparse camera control, ensuring that it can achieve even greater accuracy in interpolating camera poses.

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
