# OpenReview forum: "Boosting Camera Motion Control for Video Diffusion Transformers"
_ICLR.cc/2025/Conference — Submitted to ICLR 2025_

### Official Review · Reviewer_BUdc · 2024-10-21

**Soundness:** 2
**Presentation:** 2
**Contribution:** 2
**Rating:** 6
**Confidence:** 5

**Summary:**

This paper focuses on the challenge of camera pose control in video generation on transformer-based models (DiT). The study finds that conditioning methods, rather than pose representation, are key to performance. The authors introduce Camera Motion Guidance (CMG) to improve accuracy and a sparse camera control pipeline for easier pose specification. Their solutions work for both U-Net and DiT architectures, improving camera control across models.

**Strengths:**

1. The proposed CMG for camera motion control is clear and reasonable.

2. The experiments are comprehensive, featuring a variety of qualitative and quantitative results.

3. The paper is well-written and easy to understand.

**Weaknesses:**

1. The main insight of this paper is the significant role played by the high dimensionality of the conditioning. However, this conclusion is not clearly supported by Table 1. Without CMG, both MotionCtrl [1] (low dimension) and CameraCtrl [2] (high dimension) show poor performance, suggesting that it is specifically the combination of CMG and high-dimensional conditioning that leads to substantial improvements. The reason why DiT-CameraCtrl alone fails to work effectively remains insufficiently explained.

2. Another key insight is the design of CMG. While reasonable, it is not necessarily designed for DiT (it also has the potential for U-Net), making the connection to the main topic less coherent.

3. The motivation for sparse camera control is not well established. Unlike other sparse conditions, such as sparse image control (e.g., SparseCtrl [3]), sparse camera motion can be easily interpolated from intermediate camera poses, reducing the need for such a method.

**Questions:**

Please see the weakness.

**Details Of Ethics Concerns:**

No ethics concerns.

---

> ### Author Response · Authors · 2024-11-22
>
> ## 1. Unclear role of conditioning dimension and CMG.
> The architectural shift from U-Net to DiT introduces significant changes with multiple factors influencing performance. To better understand these effects, we conducted a series of experiments to isolate and analyze their impact.
> - We established that higher camera embedding dimension leads to better result, as shown in Table 2d vs. 1b for DiT-CameraCtrl using the same RT representation and without CMG.  Conditon-to-channel ratio (CCR) has now been added to Tables 1 and 2 to further clarify this relationship, highlighting that higher conditioning dimensions yield better outcomes.
> - DiT-CameraCtrl is actually working despite not as well without CMG, as compared to DiT-MotionCtrl which practically has no control over camera pose.  In fact, DiT-CameraCtrl achieves controllability (Table 1: 2c, rotation and translation errors) close to its U-Net counterpart (Table 1: 2a), aside from a much lower motion magnitude.
>
> We verified that increasing the conditioning dimension and introducing CMG independently contribute to performance improvements. Although the exact cause of the limited motion in DiTs remains unclear, our proposed CMG, combined with higher conditioning dimensions, achieves a motion magnitude boost of over 400% (Table 1: 2c, Supplementary Video 1) while simultaneously enhancing controllability by reducing rotation and translation errors.
>
> ## 2. CMG is not for DiT only, it has the potential for U-Net
> Indeed, it is a potential strength of our method. The proposal of CMG was motivated by challenges we observed in DiT, which is why the paper title reflects this focus. Our experiments in Table 1 suggest that existing camera conditioned schemes already yield reasonably effective controls in U-Net, but they are not as successful when implemented in DiT, which is why we focus on DiT. We observed that the degradation of camera control could be due to the overall less pronounced motion generated by OpenSora1.0 (both 1b and 2b in Table 1 have little motion). Our CMG effectively enhances the motion as pointed out in Table 1 and Figure 3, in turn enabling better camera control. While being conceptually applicable in both architectures, we focus on the challenging case (DiT) first and consider the U-Net experiments future work.
>
> ## 3. What is the purpose of sparse camera control?
> Sparse camera control has practical applications in streamlining the video creation process. For instance, this approach allows directors to focus on defining key moments, such as the starting and ending camera poses, while the model intelligently interpolates the intermediate poses to create smooth and coherent camera motion. This enables creative professionals to concentrate on high-level artistic direction rather than intricate technical details.

---

> > ### Comment · Reviewer_BUdc · 2024-11-24
> >
> > Thank you for the answers.
> >
> > The responses addressing Weakness 1 and Weakness 2 have effectively resolved my concerns.
> >
> > However, my concern regarding Weakness 3 remains. While I understand the purpose of sparse camera control, my question lies in the fact that sparse camera parameters can be numerically interpreted relatively easily as a preprocessing step. This suggests that the problem is not particularly challenging for camera control.
> >
> > In total, I acknowledge that this work has its impact on the camera control community. To this end, I will raise my rating.

---

### Official Review · Reviewer_pfNw · 2024-11-02

**Soundness:** 3
**Presentation:** 3
**Contribution:** 2
**Rating:** 5
**Confidence:** 4

**Summary:**

The paper presents a systematic analysis of design choices for incorporating camera motion control into video diffusion models. Key findings include:  1) camera control performance depends more on the choice of conditioning methods rather than camera pose representations. 2) classifier-free guidance can significantly boost camera control. 3) Camera control can be applied sparsely.

**Strengths:**

The analysis of the paper is quite thorough, and the conclusion seems reasonable. It is interesting to see a head-to-head comparison of different camera pose representation and how the condition is inserted. It is also interesting to see the effect of applying CFG helps improve camera control.

**Weaknesses:**

The claim that being the “the first camera control model for space-time video diffusion transformers” is over-claimed given Banhmani et al’s paper was arxived over three months before this submission.

The analysis is not complete, first in ablations of table 1, it misses experiments of DiT-CameraCtrl with RT representation.
Second, the analysis only evaluate the effect in terms of camera control accuracy of mostly static scenes. It does not show how different settings affect overall video quality, including how it affects object motions. It would make the analysis more meaningful to provide samples presenting dynamic objects, and includes evaluation of the video quality.

While the analysis is interesting and reassuring, the paper feels more like a technical report without new method being proposed, and the conclusions are not surprising findings.

Ln 193 “Comparing with DiT, U-Net has a worse condition-to channel ratio leading to weakened control.” is there a typo? Should DiT and U-Net be exchanged order in this sentence?

**Questions:**

Could authors show additional analysis on how different settings affect the video quality? including how they may degrade dynamic object motions.

---

> ### Author Response · Authors · 2024-11-22
>
> ## Overclaiming to be the first camera control for space-time video DiT.
> Introduced in L95-100, video DiT architectures can be broadly categorized into two types: space-time (2D+1D) and spatio-temporal (3D). Space-time (2D+1D) architectures first encode individual 2D frame into single embedding, our method injects camera pose to each video frame before batching multiple frames  for temporal attention. On the other hand, spatio-temporal (3D) architectures, like VD3D uses, simultaneously encode multiple 2D images across different time slots into a single latent embedding, which lack individual image embedding for camera pose injection using methods such as CameraCtrl, MotionCtrl and ours. Hence, camera control for these 2 architectures are inherently incompatible. To sum it up,  VD3D was the first in  spatio-temporal (3D) DiT while ours was the first in space-time (2D+1D) DiT, collectively making contribution towards improving camera control for DiTs.
>
> ## Missing experiments of DiT-CameraCtrl with RT representation.
> The experiment result is already in Table 2:(2d) which shows improved result from DiT-MotionCtrl (Table 2:(1b)) by projecting the same RT representation into higher dimensional embedding. Conditon-to-camera ratio (CCR) have now been added to Tables 1 and 2 to further clarify this relationship, highlighting that higher conditioning dimensions yield better outcomes, affirming our hypothesis.
>
> ## No moving object in study.
> Thank you for the valuable feedback. The evaluation of camera pose control in our study relies on estimating camera poses from videos using structure-from-motion techniques, such as COLMAP, which was also used in the baseline methods (MotionCtrl, CameraCtrl) and concurrent works (Vd3D and Camco). Since moving objects can degrade pose estimation accuracy, the current standard practice of all aforementioned methods is to train and evaluate camera pose control separately from dynamic object motion. The primary goal of our work was to improve camera pose guidance. Although dynamic objects were not the focus of our experiments, we do not anticipate that our method negatively affects object motion, as demonstrated in "Supplementary Video 4," where we showcase video generation with both static and dynamic scenes. We acknowledge the need for a more comprehensive analysis of object motion and video quality, and we plan to improve this aspect in future work.
>
> ## How does different setting affect video quality.
> Our method finetunes only the temporal attention of DiT, hence no video quality disparity was observed for different camera representation and conditioning methods. Camera Motion Guidance (CMG) is applied only during inference, and we conducted a sweep of the CMG scale (Table 3). As expected, (L438-443) increasing the CMG scale leads to blurrier videos due to higher camera motion, resulting in a decline in FID. We found that a CMG scale between 4 and 5 achieves the optimal balance between camera control and video quality.
>
> ## Insignificant novelty and no new method being proposed.
> VD3D reported that existing camera control methods fail for spatio-temporal transformers, and our study demonstrates similar limitations for space-time transformers. Collectively, these findings confirm that existing methods are ineffective for DiT architectures. Despite trying different combination of methods and camera representation in our study, the camera controls were still suboptimal and the small motion magnitude renders DiT-CameraCtrl  inpractical for application. It only became effective with the introduction of Camera Motion Guidance (CMG) which brings 400% improvement in camera motion.
>
> In hindsight, adapting CFG for CMG might seem straightforward; however, none of the existing camera control methods (MotionCtrl, CameraCtrl, VD3D, CamCo) implemented this approach. We argue that being able to fully lock the viewpoint is also one important aspect of camera control, which is often neglected. We confirmed methods above often fail to do so when inputing static cameras, making them not very useful in practical video editing workfows. Our CMG explictily teaches the network when to keep the camera still during training. While it is simple in design, its impact is significant, representing an original contribution in guiding camera motion in video generation.
>
> ## Is there a typo in L193? Should DiT and U-Net be exchanged order in this sentence?
> Thanks for pointing out, we have fixed that in the revised paper.

---

### Official Review · Reviewer_gvas · 2024-11-03

**Soundness:** 2
**Presentation:** 3
**Contribution:** 2
**Rating:** 5
**Confidence:** 3

**Summary:**

This paper investigates the role of camera motion control in transformer-based video diffusion models, with a particular focus on Dit-based video models. The authors identify that the dimensionality of camera control embeddings significantly impacts model performance. To enhance generative control over video outputs, they introduce a Camera Motion Guidance mechanism, which improves the model's adherence to specified motion conditions. Additionally, they propose a data augmentation technique that selectively drops some camera motion signals during training, equipping the model with robust sparse camera control capabilities. Experiments on the RealEstate10K test set demonstrate the effectiveness of these proposed methods, showing improved performance in generating realistic videos with controlled camera movement.

**Strengths:**

- The paper is well-written, with clear and structured explanations that make the content accessible and easy to follow.
- The proposed method is theoretically sound, and the authors provide a well-reasoned justification for each component.
- The authors conduct extensive experiments that rigorously explore the effects of motion control on Dit-based video generation models, offering a comprehensive evaluation of their method's effectiveness.

**Weaknesses:**

- The paper has a noticeable similarity to [1], which also examines camera motion control in transformer-based video generation models. A more detailed discussion and comparison with this related work should be provided to highlight the novel aspects of the proposed approach and clarify its distinct contributions. Additionally, experimental comparisons with VD3D should be included to further validate the effectiveness of the proposed method.
- From my perspective, the Camera Motion Guidance mechanism does not represent a significant novelty, as classifier-free guidance is already a standard setting in existing generative modeling repositories, where it can be straightforwardly adapted by substituting text conditions with camera motion conditions.

References:

[1]. VD3D: Taming Large Video Diffusion Transformers for 3D Camera Control, arXiv 2024.

**Questions:**

- What is the practical advantage of using sparse motion conditions? In Table 4, it appears that increasing the sparsity of camera motion negatively impacts inference performance. Additionally, during inference time, why not directly interpolate camera poses from the sparse keyframes and provide both the interpolated poses and key poses to the model?

---

> ### Author Response · Authors · 2024-11-22
>
> ## Similarity to VD3D and no experimental comparison
> The concurrent work VD3D based on a proprietary video model with non-standard DiT architecture and did not release any code, rendering reproduction and experimental comparisons unfeasible.
>
> Introduced in L95-100, video DiT architectures can be broadly categorized into two types: space-time (2D+1D) and spatio-temporal (3D). Space-time (2D+1D) architectures first encode individual 2D frame into single embedding, our method injects camera pose to each video frame before batching multiple frames  for temporal attention. On the other hand, spatio-temporal (3D) architectures, like VD3D uses, simultaneously encode multiple 2D images across different time slots into a single latent embedding, which lack individual image embedding for camera pose injection using methods such as CameraCtrl, MotionCtrl and ours. Hence, camera control for these 2 architectures are inherently incompatible.
>
> ## Insignificant novelty.
> VD3D reported that existing camera control methods fail for spatio-temporal transformers, and our study demonstrates similar limitations for space-time transformers. Collectively, these findings confirm that existing methods are ineffective for DiT architectures. Despite trying different combination of methods and camera representation in our study, the camera controls were still suboptimal and the small motion magnitude renders DiT-CameraCtrl  inpractical for application. It only became effective with the introduction of Camera Motion Guidance (CMG) which brings 400% improvement in camera motion.
>
> In hindsight, adapting CFG for CMG might seem straightforward; however, none of the existing camera control methods (MotionCtrl, CameraCtrl, VD3D, CamCo) implemented this approach to guide or improve camera motion control. On the other hand, we argue that being able to fully lock the viewpoint is also one important aspect of camera control, which is often neglected. We confirmed methods above often fail to do so when inputting static cameras, making them not very useful in practical video editing workfows. Our data augmentation and CMG explicitly teaches the network when to keep the camera still during training. While it is simple in design, CMG's impact is significant, representing an original contribution in guiding camera motion in video generation.
>
> ## What is the purpose of sparse camera control?
> Sparse camera control has practical applications in streamlining the video creation process. For instance, this approach allows directors to focus on defining key moments, such as the starting and ending camera poses, while the model intelligently interpolates the intermediate poses to create smooth and coherent camera motion. This enables creative professionals to concentrate on high-level artistic direction rather than intricate technical details.  Table 4 demonstrates that utilizing sparse camera poses in conjunction with our data augmentation pipeline results in only modest degradation in camera accuracy. This promising outcome lays the groundwork for future research aimed at further enhancing sparse camera control methods.

---

> > ### Comment · Reviewer_gvas · 2024-11-26
> > **Official Response by Reviewer gvas**
> >
> > Thank you for the rebuttal. I still have a question about sparse camera control. In practical scenarios, providing dense camera poses allows for more flexible control of video generation. Even with only the start and end camera poses, intermediate poses can be pre-interpolated and fed into the models. Is there any quantitative or qualitative comparison between the proposed sparse camera control and this simple baseline?

---

### Official Review · Reviewer_czWF · 2024-11-03

**Soundness:** 3
**Presentation:** 3
**Contribution:** 3
**Rating:** 6
**Confidence:** 5

**Summary:**

This paper provides in-depth analysis on the design choices of adding camera control to U-Net-based and DiT-based video generation models. The investigated choices include the representation of camera parameters, camera conditioning methods, as well as guidance methods of diffusion model. This paper reveals that it is the conditioning methods more crucial than the representation, which is opposite to what is commonly believed. In terms of guidance methods, authors propose a camera motion guidance (CMG) that introduces camera motion guidance in the classifier-free guidance. The integration of these findings results in a camera control method with significant improvement in terms of motion metric.

**Strengths:**

Overall I think this is a good paper. Although technical novelty is limited when viewed in isolation, technical improvements come from an inspiring analysis of the details, and significant improvements are observed in experiments.
- Paper is easy to follow
- The in-depth analysis is informative and inspiring. I especially like the concept of condition-to-ratio. And the subsequently emphasizing the condition method rather than camera embedding looks reasonable.
- The design of camera motion guidance looks reasonable as well.
- Experiment results in table 1 are encouraging, where DiT-based method matches the performance of U-Net-based method.

**Weaknesses:**

- The idea of condition-to-ratio should be further investigated or emphasized, such as adjusting the number of dimensions to vary condition-to-ratio so that its effect can be observed more clearly.
- While authors state the proposed solution is broadly applicable across DiT models and U-Net models, I only see results of DiT models in experiments. U-Net models with CMG should be included to verify the claim.
- Only one backbone DiT model is evaluated
- In terms of sparse camera control, I wonder how to obtain the sparse camera control signal with good naturalness and smoothness. A way I can think of is subsampling from **a complete one**, but in such cases we can just input the complete signal instead without the need of using a sparse camera control. Any thoughts?

**Questions:**

See weaknesses.

---

> ### Author Response · Authors · 2024-11-22
>
> ## Adjusting dimensions to vary condition-to-channel ratio (CCR).
> DiT-CameraCtrl encodes the camera pose to match the embedding dimensions of DiT, ensuring a fixed 1:1 CCR that cannot be adjusted. In contrast, using the DiT-MotionCtrl method of concatenation inflates the input channel dimension across all linear layers, significantly increasing GPU memory usage and risking out-of-memory errors. In Table 2, even without CMG or the "better" Plücker coordinate, we demonstrate that projecting RT camera poses into higher-dimensional embeddings improves the results for baselines (Table 2: 2d vs. 1b). To further clarify this relationship, we have now added CCR to Tables 1 and 2, highlighting that higher conditioning dimensions yield better outcomes.
>
> ## Not tested on U-Net
> Our experiments in Table 1 suggest that existing camera conditioned schemes already yield reasonably effective controls in U-Net, but they are not as successful when implemented in DiT, which is why we focus on DiT. We observed that the degradation of camera control could be due to the overall less pronounced motion generated by OpenSora1.0 (both 1b and 2b in Table 1 have little motion). Our CMG effectively enhances the motion as pointed out in Table 1 and Figure 3, in turn enabling better camera control. While being conceptually applicable in both architectures, we focus on the challenging case (DiT) first and consider the U-Net experiments future work. However, we agree with the reviewer that the applicability on U-Net remains to be confirmed so we have removed the related statement in the revised paper.
>
> ## Only one backbone DiT model is evaluated.
> At the time of our project, we only found two open-source space-time (2D+1D) video DiT models for experiments: OpenSora, which we used, and Latte. OpenSora adopts a generic DiT architecture from Latte, making our analysis broadly applicable to both and to other standard DiT video models. Another open-source model, OpenSora-Plan, employs a spatio-temporal (3D) architecture hence unsuitable. VD3D, a concurrent camera control work based on  spatio-temporal DiT,  evaluated their camera control method on a closed video model, SnapVideo with non-standard DiT architecture which is not released.
>
> ## How to obtain sparse pose?
>
> In training, we randomly drop camera poses from some frames (L287-288), making the model to learn to interpolate the missing poses to generate smooth motion. In inference, we subsample evenly from the complete poses. Table below shows the frame index (out of 16 frames) used for each sparsity ratio in Table 4.
>
> | Sparsity ratio | frame index used in inference |
> | :--------------: | :-----------------------------: |
> | 0% | 0, 1, 2, 3, 4, 5, 6, 7, 8, 9, 10, 11, 12, 13, 14, 15  |
> | 47% | 0, 2, 4, 6, 8, 10, 12, 14, 15 |
> | 73% | 0, 4, 8, 12, 15 |
> | 87% | 0, 8, 15 |
> | 93% | 0, 15 |
>
> ## What is the purpose of sparse camera control?
> Sparse camera control has practical applications in streamlining the video creation process. For instance, this approach allows directors to focus on defining key moments, such as the starting and ending camera poses, while the model intelligently interpolates the intermediate poses to create smooth and coherent camera motion. This enables creative professionals to concentrate on high-level artistic direction rather than intricate technical details.

---

> > ### Comment · Reviewer_czWF · 2024-11-26
> > **Official Response of Reviewer czWF**
> >
> > Thanks authors for the feedback. My concerns in terms of only one DiT backbone and no test on U-Net are addressed.
> >
> > As for my question on sparse camera control, my question is the sparse camera control signals are obtained from complete ones as authors explained in the response, "In inference, we subsample evenly from the complete poses." If we already have complete camera control signals, why don't we just take the complete ones as condition?
> >
> > In the response to "What is the purpose of sparse camera control?", authors stated "this allows directors to focus on defining key moments", I agree but I think it's hard for directors to define only camera at sparse key moments. In my opinion, directors need to always define the complete camera trajectory then subsampling from it to obtain the sparse camera control. Could authors discuss this point?
> >
> > Overall, I still think this work has its value so I keep my rating.

---

### Author Response · Authors · 2024-12-03
**Sparse Camera Control**

Dear reviewers:


> **Reviewer czWF**:  If we already have complete camera control signals, why don't we just take the complete ones as condition?
>
> **Reviewer gvas and BUdc**: Why not interpolate the camera poses?

We evaluated our method using sparse camera poses derived from complete pose in the dataset, which, in practice, may not always be immediately available during real-time filming.

Thanks reviewers for the insightful comment, intermediate camera poses can be generated from keyframes using interpolation. We conducted additional experiments to test this approach by inputting linearly interpolated camera poses. As shown in the table below, the results reveal only a minor difference in camera control accuracy compared to directly using sparse camera poses, provided the sparsity remains below 90%. Given the small timestep intervals (16 frames in 2 seconds) in our experiments, linear interpolation proved effective. However, for longer intervals between keyframes, more sophisticated interpolation techniques may need to be considered.

Professional tools such as After Effects and Maya already offer camera keyframe functionalities. We believe our sparse camera control method could be naturally integrated into these tools, which further streamlines the process by potentially eliminating the need for interpolation altogether, offering a simpler and more efficient solution for video content creation workflows.

| # Frames dropped | Sparsity ratio | Rot error (sparse)| Rot error (interpolated)| Transl error (sparse) | Transl error (interpolated)|
| :--:| :--: | :--: | :--: | :--: | :--: |
| 7 |      47%       |  0.181  | 0.187 | 0.627| 0.615 |
| 11 |      73%       |  0.203  | 0.194 |0.650| 0.663 |
| 13 |      87%       |  0.218  | 0.212 |0.755| 0.713 |
|14 |      93%       |  0.291  | 0.233 |0.768| 0.762 |

---

### Meta-Review · Area_Chair_g8dX · 2024-12-17

**Metareview:**

The reviewers generally appreciated the paper’s systematic analysis of camera control in video diffusion models, highlighting the insightful exploration of conditioning methods and the introduction of Camera Motion Guidance (CMG). However, concerns were about the limited novelty and application, and insufficient experimental evaluation. Specifically, the paper is not convincing in validating its approach across U-Net models, and broader video quality metrics. Additionally, the justification for sparse camera control remains unconvincing given the ease of interpolation. I’d recommend the authors to consider all these and further improve the work.

**Additional Comments On Reviewer Discussion:**

There are discussions about reviewer's concern, focusing on: 1) the justification and practicality of sparse camera control, 2) the novelty of the approach compared to existing methods, and 3) the lack of ablations on output video quality. Overall, the paper's novelty is somewhat limited, and questions regarding the practical use cases and advantages of sparse camera control remain unresolved.

---

### Decision · Program_Chairs · 2025-01-22

Reject